# Lower Free Thyroxine Levels Are Associated with Diabetic Kidney Disease in Males with Type 2 Diabetes Mellitus: An Observational Cross-Sectional Study

**DOI:** 10.3390/biomedicines12102370

**Published:** 2024-10-17

**Authors:** Jianan Shang, Yixuan Zheng, Meng Zhang, Meng Li, Wei Qiang, Jing Sui, Hui Guo, Bingyin Shi, Mingqian He

**Affiliations:** 1Department of Ultrasound, The First Affiliated Hospital of Xi’an JiaoTong University, 277 West Yanta Road, Xi’an 710061, China; sjnecho@xjtufh.edu.cn; 2Department of Endocrinology, The First Affiliated Hospital of Xi’an JiaoTong University, 277 West Yanta Road, Xi’an 710061, China; zhengyx0921@163.com (Y.Z.); mengbaba19921025@163.com (M.Z.); limeng2000@126.com (M.L.); weiqiang@xjtufh.edu.cn (W.Q.); suijing1029@163.com (J.S.); guohui0709@mail.xjtu.edu.cn (H.G.); shibingy@126.com (B.S.)

**Keywords:** diabetic kidney disease, thyroid function, free thyroxine, type 2 diabetes mellitus

## Abstract

**Objectives**: We aimed to explore the correlation between thyroid function and diabetic kidney disease (DKD) in patients with type 2 diabetes mellitus (T2DM). **Methods**: A total of 7516 T2DM patients were enrolled and grouped according to DKD status. Clinical parameters, including blood glucose parameters, thyroid function, and indicators of renal impairment, were collected and compared between the DKD and Non-DKD groups. Correlation analysis and univariate/multivariate logistic regression analyses were performed. **Results**: Age, T2DM duration, the use of insulin and lipid-lowering drugs, systolic and diastolic blood pressure, body mass index, and fasting blood glucose levels were greater in the DKD group than in the Non-DKD group (*p* < 0.001). Notably, compared with those in the Non-DKD group, patients in the DKD group had lower triiodothyronine (T3), thyroxine (T4), free triiodothyronine (FT3), and free thyroxine (FT4) levels and higher thyrotropin levels (*p* < 0.001). Univariate logistic regression analysis revealed that T3, T4, FT3, and FT4 levels were negatively correlated with the risk of DKD. Spearman correlation analysis confirmed that T3, T4, FT3, and FT4 levels were negatively correlated with blood urea nitrogen levels, blood creatinine levels, and the urinary albumin-to-creatinine ratio (*p* < 0.05). Multivariate logistic regression analysis revealed that a greater FT4 level was a protective factor against DKD in T2DM patients, especially in males, with a cut-off value of 13.35 pmol/L (area under the curve = 0.604). **Conclusions**: Thyroid hormone levels, especially FT4 levels, were significantly negatively correlated with DKD in T2DM patients.

## 1. Introduction

Diabetic nephropathy, clinically known as diabetic kidney disease (DKD), is one of the most common microvascular diseases in diabetes patients, affecting approximately 20–40% of diabetes patients worldwide [1]. DKD has become the leading cause of end-stage renal disease, which is closely related to increased rates of disability and mortality in diabetes patients [1]. The pathological changes in the kidneys in the early stage of DKD can be reversed. However, once DKD progresses to end-stage renal disease, the condition is irreversible, and patients must eventually receive life-sustaining treatments such as dialysis or kidney transplantation; these treatments cause serious economic losses to society and families and seriously affect patients’ quality of life [2]. Therefore, exploring potential factors that influence the pathogenesis of DKD is highly important for prevention and early diagnosis to improve patient prognosis and quality of life.

The pathogenesis of DKD has not yet been fully elucidated. Previous studies have shown that hypertension, hyperglycemia, dyslipidemia, and other factors are involved in the process of DKD development [3,4]. Multiple clinical studies have shown that abnormal thyroid function may affect glucose metabolism, insulin sensitivity, and the development of long-term diabetes complications [5,6]. Diabetes patients with hypothyroidism are at greater risk of developing DKD and diabetic retinopathy [7,8]. Recent studies have reported that the occurrence of DKD in euthyroid diabetes patients is also correlated with thyroid function [9,10].

To elucidate the correlation between thyroid function and the pathogenesis of DKD, we conducted a cross-sectional study in which more than 7000 patients with type 2 diabetes mellitus (T2DM) were enrolled and grouped on the basis of the presence of DKD. We hypothesized that thyroid function may be significantly associated with DKD in patients with T2DM.

## 2. Materials and Methods

### 2.1. Patient Inclusion

This study was approved by the Institutional Review Board of the First Affiliated Hospital of Xi’an JiaoTong University (XJTU1AF2018LSK-055). Patients who were diagnosed with T2DM and who had been hospitalized in the Department of Endocrinology at the First Affiliated Hospital of Xi’an Jiaotong University from January 2013 to December 2022 were included. The exclusion criteria were as follows: gestational diabetes; type 1 diabetes mellitus or other special types of diabetes; T2DM combined with stress; chronic nephritis; primary glomerulonephritis; urinary tract infection; urinary tract stones or tumors caused by acute infection and other factors; severe liver dysfunction; malignant tumors; connective tissue disease; or autoimmune diseases.

All patients were evaluated by professional clinicians, and the presence of DKD was determined according to the following international diagnostic standards: persistently decreased estimated glomerular filtration rate (eGFR) < 60 mL/min/1.73 m^2^ and/or elevated urinary albumin-to-creatinine ratio (UACR) > 30 mg/gCr [11,12]. The eGFR was estimated via the Cockcroft–Gault and Chronic Kidney Disease Epidemiology Collaboration (CKD–EPI) equations [13]. All patients were staged via the Kidney Disease: Improving Global Outcomes (KDIGO) consensus, which uses the eGFR and UACR [14]. A total of 7516 patients were ultimately included in this study, and all patients provided written informed consent.

### 2.2. Research Methods

The patients’ basic information, medication history, anthropometric indicators, laboratory examination results, and ultrasound measurements were collected from the electronic medical records system. Basic information included age, sex, and T2DM duration. Medication history included the use of insulin, oral hypoglycemic drugs, glucagon-like peptide-1 analogues, and lipid-lowering drugs. The anthropometric indicators included body mass index (BMI), systolic blood pressure (SBP), and diastolic blood pressure (DBP). The laboratory examination results included glycated hemoglobin levels, renal function, urine test results, and thyroid function data. Ultrasound measurements included the thickness of the thyroid isthmus measured by professional medical sonographers.

### 2.3. Laboratory Measurements

Venous blood samples were collected in the morning after an 8 h fast, and urine samples included both fasting urine samples collected in the morning and 24 h total urine samples.

Fasting blood glucose (FBG), glycated albumin, total cholesterol (TC), triglycerides (TGs), high-density lipoprotein cholesterol (HDL-c), low-density lipoprotein cholesterol (LDL-c), and serum albumin (ALB) were tested via the LABOSPECT 008AS automatic biochemical analyzer (HITACHI, Tokyo, Japan) using standard reagents to the nearest 0.01 mmol/L, 0.1%, 0.01 mmol/L, 0.01 mmol/L, 0.01 mmol/L, 0.01 mmol/L, and 0.01 mmol/L. Glycated hemoglobin (HbA1c) was measured to the nearest 0.1% using an HLC-723G8 automatic analyzer (TOSOH BIOSCIENCE, Tokyo, Japan).

Blood urea nitrogen (BUN), serum creatinine, cystatin C (Cys-C), uric acid (UA), urinary creatinine, microalbumin, 24 h microalbumin, and 24 h urine protein (24hU-TP) levels were measured via a LABOSPECT 008AS automatic biochemical analyzer (HITACHI, Tokyo, Japan) with standard reagents to the nearest 0.01 mmol/L, 1 µmol/L, 0.01 mg/L, 0.1 µmol/L, 1 µmol/L, 0.01 mg/L, 0.01 mg/24 h, and 0.01 g/24 h, respectively.

Free triiodothyronine (FT3), free thyroxine (FT4), thyrotropin (TSH), triiodothyronine (T3), thyroxine (T4), and thyroid peroxidase antibody (TPOAB) were measured by radioimmunoassay via GC-2016 16th automatic sampling change equipment (ZONKIA, Anhui, China) with standard reagents to the nearest 0.01 pmol/L, 0.01 pmol/L, 0.01 µIU/mL, 0.01 ng/mL, 0.01 µg/dL, and 1 U/mL, respectively. Thyroglobulin antibody (TGAB) and thyroid microsomal antibody (TMAB) were measured by radioimmunoassay via an FM-2000 gamma immune counter (KAIPU, Xi’an, China) to the nearest 0.01% and 0.01%, respectively.

### 2.4. Statistical Analysis

Normally distributed variables are presented as the means ± standard deviations, and between-group comparisons were performed via independent-sample *t* tests. Abnormally distributed variables are reported as medians and interquartile ranges, and the between-group comparisons were conducted via the Mann–Whitney U test. The categorical variables are expressed as numbers and percentages, and the between-group comparisons were performed with chi-square tests. Spearman correlation analysis was performed to explore the correlations between thyroid function and related indicators of renal impairment. Univariate/multivariate logistic regression was conducted to evaluate the factors influencing DKD in patients with T2DM. Ordered logistic regression was performed to explore the association between thyroid hormone levels and DKD stages. A two-tailed *p* < 0.05 was considered to indicate statistical significance. SPSS 25.0 software (IBM, Chicago, IL, USA) was used for statistical analysis.

## 3. Results

### 3.1. Clinical Characteristics of the Participants

As Figure 1 shows, a total of 7516 T2DM patients, including 5549 DKD patients and 1967 Non-DKD patients, were enrolled in the study. Table 1 shows the basic clinical information of the patients in the Non-DKD and DKD groups. The age and T2DM duration of the DKD patients were significantly greater than those of the Non-DKD patients (*p* < 0.001). The percentage of individuals with insulin and lipid-lowering drug use was greater in the DKD group than in the Non-DKD group, whereas the Non-DKD group had a higher percentage of individuals with oral hypoglycemic drug use (*p* < 0.001). Compared with the Non-DKD group, the DKD group presented higher SBP, DBP, BMI, FBG, TC, TG, and LDL-c levels. The UA, Cys-C, blood creatinine, BUN, urinary macroglobulin, and 24hU-TP levels and UACR in the DKD group were greater than those in the Non-DKD group, whereas the ALB, eGFR, and urinary creatinine levels in the DKD group were lower than those in the Non-DKD group (*p* < 0.001).

Compared with the Non-DKD group, the DKD group presented lower levels of T3 (1.05 ± 0.39 ng/mL vs. 1.14 ± 0.41 ng/mL), T4 (7.17 ± 2.26 µg/dL vs. 7.51 ± 2.52 µg/dL), FT3 (4.16 ± 1.57 pmol/L vs. 4.58 ± 1.57 pmol/L), and FT4 (14.27 ± 4.15 pmol/L vs. 15.47 ± 5.32 pmol/L), and higher levels of TSH (1.96 [2.18] µIU/mL vs. 1.74 [1.78] µIU/mL) (all *p* < 0.001). There were no significant between-group differences in the thyroid isthmus thickness or TGAB, TMAB, and TPOAB levels.

### 3.2. Univariate Analysis of the Effect of Thyroid Function on the Risk of DKD

Univariate logistic regression analysis revealed that thyroid hormones, including T3 (odds ratio (OR): 0.469, 95% confidence interval (CI): 0.381–0.577), T4 (OR: 0.940, 95% CI: 0.913–0.969), FT3 (OR: 0.789, 95% CI: 0.746–0.835), and FT4 (OR: 0.926, 95% CI: 0.908–0.944), were negatively correlated with the risk of DKD (Figure 2). The levels of TSH, TGAb, TMAb, and TPOAb were not significantly correlated with the risk of DKD (*p* > 0.05).

### 3.3. Correlation of Thyroid Function with Indicators of Renal Impairment

Next, we performed Spearman correlation analysis to investigate the relationships between thyroid function and indicators of renal impairment (Figure 3 and Appendix A) and found that T3, T4, FT3, and FT4 levels were negatively correlated with Cys-C, blood creatinine, BUN, urinary creatinine, urinary microglobulin, and 24hU-TP levels and positively correlated with the urinary creatinine level and the eGFR. In addition, T3, T4, FT3, and FT4 levels were negatively correlated with glycated albumin, HbA1c, and FBG levels and the duration of T2DM.

### 3.4. Multivariate Analysis of the Independent Effect of Thyroid Function on the Risk of DKD

To control for confounding factors, we established three multivariate logistic regression models via the forward (conditional) stepwise regression method to explore the effects of thyroid hormones on the risk of DKD. In Model 1, age, sex, and BMI were adjusted for; in Model 2, age, sex, BMI, T2DM duration, and HbA1c levels were adjusted for; and in Model 3, age, sex, BMI, T2DM duration, HbA1c, SBP, DBP, UA, TC, TG, HDL-c, LDL-c, and ALB levels were adjusted for.

Table 2 shows the effects of thyroid hormones, including T3, T4, FT3, and FT4, on the risk of DKD in the three models. We also found that lower thyroid hormone levels, including T3, T4, FT3, and FT4, were associated with higher DKD stages (Appendix A). In Model 3, after adjusting for multiple factors, the FT4 level was still significantly negatively correlated with the risk of DKD, with an OR of 0.963 and 95% CI of 0.934–0.993.

### 3.5. Receiver Operating Characteristic Analysis to Determine the Cut-Off Value of FT4 Levels

Given that a lower FT4 level is an independent risk factor for DKD, we conducted a receiver operating characteristic (ROC) analysis on patients grouped by sex (Figure 4). The results revealed that the effect of the FT4 level on the risk of DKD was more significant in males, with a cut-off value of 13.35 pmol/L (the Youden index was 0.166, the sensitivity was 74.0%, and the specificity was 57.4%), and the area under the curve (AUC) was 0.604 (*p* < 0.001).

## 4. Discussion

This cross-sectional study involving more than 7000 T2DM patients revealed that thyroid function was significantly negatively correlated with the prevalence of DKD and that the FT4 level was an independent protective factor against DKD. In addition, an FT4 concentration <13.35 pmol/L could be used as a potential predictor of DKD in male patients with T2DM. This large clinical study provides a theoretical basis and clinical reference for the long-term management of thyroid function in T2DM patients.

In this study, T3, T4, FT3, and FT4 levels were negatively correlated with the prevalence and stages of DKD, especially the FT4 level, which was an independent protective factor against DKD. Previous clinical studies have shown that increased FT3 levels and decreased TSH levels are protective factors against DKD [6,9,10] and are associated with the prevalence of moderate-to-severe DKD [8] after confounding factors are excluded. However, the relationship between FT4 levels and the risk of DKD is not clear. While patients who had drugs that may affect thyroid function were not excluded in this study, our findings are consistent with a cross-sectional study involving 248 euthyroid T2DM patients which revealed that the FT4 level was negatively correlated with microvascular complications, including DKD [10]. In contrast, another cross-sectional study including more than 4000 patients revealed that the FT4 level was positively correlated with DKD in T2DM patients [15]. This contradiction may be attributed to differences in research methods, sample sizes, and patient sources. As a large clinical study involving more than 7000 patients, our results revealed that a greater FT4 level was an independent protective factor against DKD in T2DM patients.

Thyroid dysfunction, especially low thyroid function, is closely linked to multiple metabolic disorders, such as hyperglycemia [16], metabolic dysfunction-associated fatty liver disease [17], and dyslipidemia [18], which may be attributed to the vital role of thyroid hormones in energy, glucose, and lipid metabolism [19,20]. We found that thyroid function is significantly associated with glycemia parameters and renal function (Figure 3), which are commonly involved in the pathogenesis of not only T2DM and DKD but also other metabolic disorders, such as cardiovascular disease, nonalcoholic fatty liver disease, and chronic kidney disease [21,22,23,24], indicating the underlying mechanism between thyroid dysfunction and metabolic abnormalities.

We conducted ROC analysis to obtain the cut-off value of 13.35 pmol/L for the FT4 concentration, which was higher than the previously reported value of 9.69 nmol/L in a cross-sectional study involving 248 euthyroid patients with T2DM [10]. This discrepancy may be related to the differences in sex composition, sample size, thyroid function, and population source between the two studies. Considering the heavy economic burden of DKD on patients and society, the FT4 cut-off value of 13.35 pmol/L obtained in this large observational study may be important for DKD screening in male T2DM patients.

Mechanistically, thyroid hormones may participate in the pathogenesis of DKD through multiple pathways. First, alterations in the circulating thyroid hormone concentration can regulate the function of thyroid hormone receptors, which are located on the surface of vascular endothelial cells [25]. Studies have shown that reduced nitric oxide (NO) availability in endothelial cells can lead to endothelial dysfunction, which in turn leads to changes in kidney hemodynamics and glomerular filtration [26,27,28,29], whereas thyroid hormones can regulate NO production in vascular endothelial cells [30]. Therefore, NO production in vascular endothelial cells may be the direct pathway through which thyroid hormones are involved in multiple vascular complications, including DKD [10]. Further in vivo and in vitro experiments are needed to investigate the underlying mechanism of the effects of thyroid hormones on DKD risk.

As important hormones involved in the regulation of metabolism and other biochemical processes, thyroid hormones may be involved in the pathogenesis of DKD through multiple biological changes. Previous studies have shown that insulin secretion-associated hyperglycemia can be involved in the pathogenesis of DKD via multiple pathways including the polyol pathway [31,32]. Thyroid hormones, especially FT3 and FT4, can act as insulin agonists or antagonists in different organs [33], and several studies have confirmed that hypothyroidism is associated with the occurrence of insulin resistance [34,35]. In addition, oxidative stress may also be one of the pathways by which thyroid hormones are involved in the pathogenesis of DKD. Insulin resistance can lead to oxidative stress, and hypothyroidism is closely related to the upregulation of oxidative stress [36,37]. A study on thyroid hormone replacement therapy confirmed that thyroid hormone supplementation could slow the progression of DKD by downregulating oxidative stress [38,39]. In summary, the mechanism underlying the effects of thyroid hormone levels on the occurrence and progression of DKD is still not fully understood, and further mechanistic exploration is needed.

Given the potential relationship between thyroid hormone levels and the risk of DKD, the benefit of replacement therapy for DKD patients also has guiding significance for clinical management. Currently, there are numerous clinical studies on the use of levothyroxine (LT4) for the treatment of DKD patients. A prospective clinical trial revealed that 48 weeks of LT4 supplementation could reduce the UA levels and urinary albumin excretion rate (UAER) in DKD patients with early-stage subclinical hypothyroidism [39]. Another double-blind clinical trial confirmed that LT4 supplementation could reduce the UAER in DKD patients with early-stage subclinical hypothyroidism and restore it to the rate in euthyroid patients [38]. However, another observational study failed to confirm that treatment of subclinical hypothyroidism can reduce the risk of DKD [7]. Overall, the benefit of thyroid hormone replacement therapy for patients with DKD complicated by subclinical hypothyroidism is clear, and further long-term studies are needed to investigate whether thyroid hormone replacement therapy can reduce the incidence of DKD.

This study also has several limitations. First, as a single-center, cross-sectional observational study, our study lacked data on the duration of DKD, and our results could not be used to determine the causal relationship between thyroid hormone levels and the development of DKD. Second, the patients in this study were all from Northwest China, which limits the generalizability of the findings to other populations. Additionally, our study did not focus on euthyroid T2DM patients; thus, patients who had a history of thyroid disease or drugs that may affect thyroid function were not excluded from this study, which may to some extent impact our results. Lastly, the normal range of TPOAB (<15 U/mL) may cause floor effects, which may explain the limited difference in TPOAB levels between the DKD group and the Non-DKD group.

## 5. Conclusions

Taken together, we found that thyroid hormone levels, especially the FT4 level, were significantly negatively correlated with the prevalence of DKD, which emphasized the importance of regular thyroid function tests and timely thyroid hormone replacement therapy in the long-term management of T2DM patients. Our findings also provide new ideas regarding the pathogenesis, early diagnosis, and treatment of DKD and pave the way for further research on the pathogenesis of DKD and clinical management strategies for DKD.

## Figures and Tables

**Figure 1 biomedicines-12-02370-f001:**
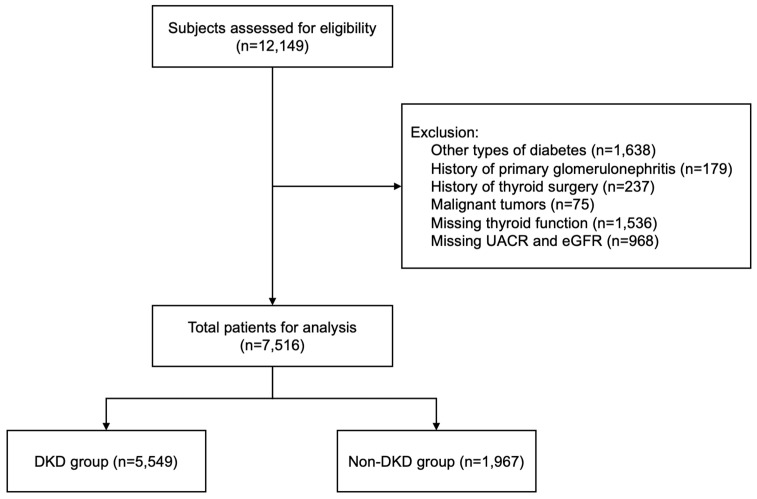
Flowchart of the inclusion and exclusion of participants. UACR, urinary albumin-to-creatinine ratio; eGFR, estimated glomerular filtration rate; DKD, diabetic kidney disease.

**Figure 2 biomedicines-12-02370-f002:**
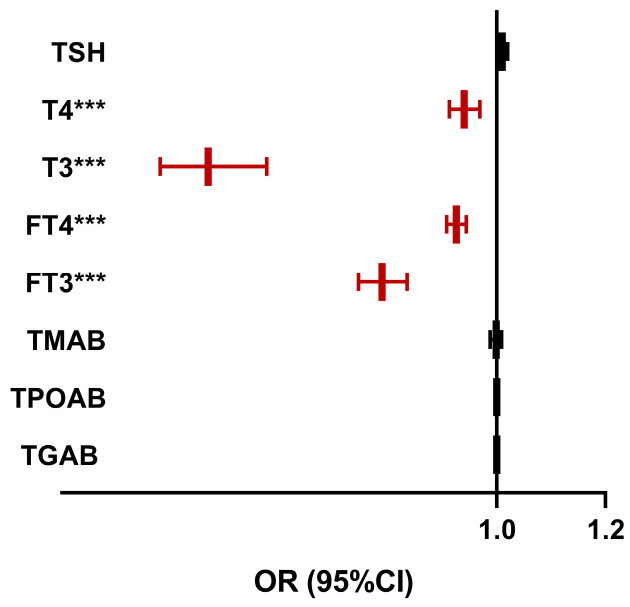
Forest graph showing the results of univariate logistic regression analyses of the association between thyroid function and diabetic kidney disease (DKD). TSH, thyrotropin; T4, thyroxine; T3, triiodothyronine; FT4, free thyroxine; FT3, free triiodothyronine; TMAB, thyroid microsomal antibody; TPOAB, thyroid peroxidase antibody; TGAB, thyroglobulin antibody; OR, odds ratio; CI, confidence interval. *** *p* < 0.001: significant correlation between thyroid function and DKD.

**Figure 3 biomedicines-12-02370-f003:**
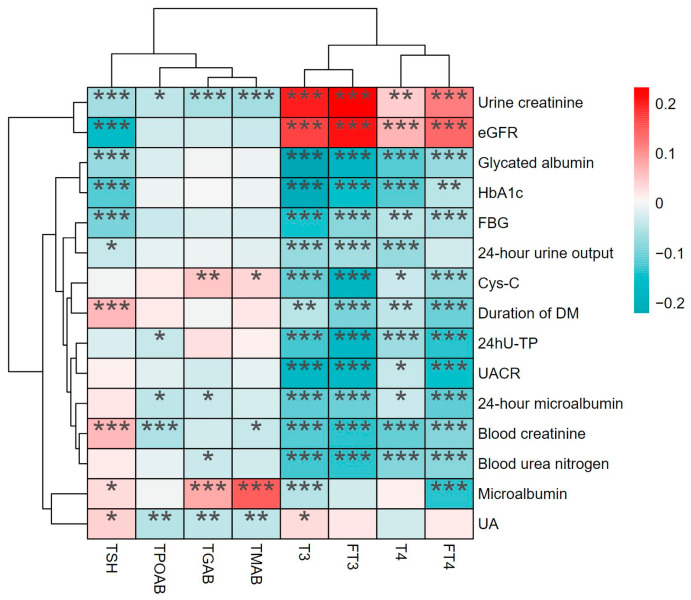
Results of Spearman’s correlation analyses of the correlations between thyroid function and DKD-related parameters. TSH, thyrotropin; T4, thyroxine; T3, triiodothyronine; FT4, free thyroxine; FT3, free triiodothyronine; TMAB, thyroid microsomal antibody; TPOAB, thyroid peroxidase antibody; TGAB, thyroglobulin antibody; eGFR, estimated glomerular filtration rate; HbA1c, glycated hemoglobin; FBG, fasting blood glucose; Cys-C, cystatin C; UACR, urinary albumin-to-creatinine ratio; 24hU-TP, 24 h urine protein. UA, uric acid. * *p* < 0.05, ** *p* < 0.01, *** *p* < 0.001: significant correlation between thyroid function and DKD-related indicators.

**Figure 4 biomedicines-12-02370-f004:**
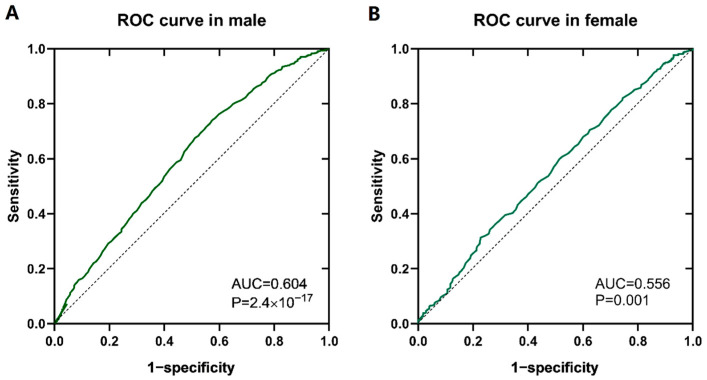
ROC analysis of the effect of the FT4 concentration on the risk of diabetic kidney disease in patients of different sexes. (**A**) ROC analysis of male patients. (**B**) ROC analysis of female patients.

**Table 1 biomedicines-12-02370-t001:** Clinical characteristics of the participants.

Variable	Non-DKD (N = 5549)	DKD (N = 1967)	*p* Value
Male, No. (%)	3451 (62.2%)	1283 (65.2%)	**0.017**
Age (years)	52.9 ± 15.6	58.8 ± 13.1	**<0.001**
T2DM duration (years)	5.0 (10.0)	11.0 (11.0)	**<0.001**
Medication usage, No. (%)			
Insulin	3473 (62.6%)	1602 (81.4%)	**<0.001**
Oral hypoglycemic drugs	3812 (68.7%)	1009 (51.3%)	**<0.001**
GLP-1 analogues	183 (3.3%)	59 (3.0%)	0.553
Lipid-lowering drugs	4071 (73.4%)	1635 (83.1%)	**<0.001**
Current smoker, No. (%)	2077 (37.4%)	817 (41.5%)	**0.001**
Current alcohol user, No. (%)	2374 (42.8%)	901 (45.8%)	**0.020**
Family history of diabetes, No. (%)	2876 (51.8%)	1073 (54.6%)	**0.038**
Subclinical hypothyroidism, No. (%)	319 (5.8%)	172 (8.7%)	**<0.001**
Body mass index (kg/m^2^)	24.24 ± 3.89	24.56 ± 3.61	**0.004**
Systolic blood pressure (mmHg)	129.38 ± 17.89	140.70 ± 21.87	**<0.001**
Diastolic blood pressure (mmHg)	78.95 ± 11.07	81.88 ± 12.47	**<0.001**
Total cholesterol (mmol/L)	4.16 ± 1.14	4.26 ± 1.39	**0.002**
Triglycerides (mmol/L)	1.40 (1.14)	1.45 (1.29)	**0.001**
HDL-c (mmol/L)	0.99 ± 0.28	0.99 ± 0.32	0.703
LDL-c (mmol/L)	2.45 ± 0.86	2.51 ± 1.06	**0.018**
Serum albumin (g/L)	39.62 ± 5.18	37.07 ± 8.04	**<0.001**
Fasting blood glucose (mmol/L)	8.39 (6.86)	9.15 (7.29)	**<0.001**
HbA1c (%)	8.93 ± 2.49	8.99 ± 2.23	0.436
Glycated albumin (%)	24.4 ± 10.3	24.4 ± 10.0	0.570
Uric acid (µmol/L)	315.8 ± 127.6	341.7 ± 159.1	**<0.001**
eGFR (mL/min/1.73 m^2^)	108.17 ± 21.03	90.83 ± 29.89	**<0.001**
Cystatin C (mg/L)	0.80 (0.29)	0.98 (0.54)	**<0.001**
Blood creatinine (µmol/L)	55 (20)	65 (37)	**<0.001**
Blood urea nitrogen (mmol/L)	5.65 ± 2.37	7.37 ± 3.99	**<0.001**
Urine creatinine (µmol/L)	7534 (7325)	5656 (5077)	**<0.001**
Microalbumin (mg/L)	1.60 (10.50)	24.1 (179.48)	**<0.001**
UACR (mg/gCr)	12.00 (14.15)	150.45 (642.99)	**<0.001**
24 h microalbumin (mg/24 h)	15.00 (16.65)	156.90 (749.30)	**<0.001**
24 h urine protein (g/24 h)	0.05 (0.05)	0.26 (1.13)	**<0.001**
24 h urine output (mL)	2000 (1250)	2000 (1200)	0.709
Triiodothyronine (T3, ng/mL)	1.14 ± 0.41	1.05 ± 0.39	**<0.001**
Thyroxine (T4, µg/dL)	7.51 ± 2.52	7.17 ± 2.26	**<0.001**
Thyrotropin (µIU/mL)	1.74 (1.78)	1.96 (2.18)	**<0.001**
Free T3 (pmol/L)	4.58 ± 1.57	4.16 ± 1.57	**<0.001**
Free T4 (pmol/L)	15.47 ± 5.32	14.27 ± 4.15	**<0.001**
Thyroglobulin antibody (%)	3.39 (2.26)	3.40 (2.01)	0.793
Thyroid microsomal antibody (%)	2.67 (1.72)	2.70 (1.61)	0.559
Thyroid peroxidase antibody (U/mL)	15.0 (0.0)	15.0 (0.0)	0.374
Thickness of the thyroid isthmus (mm)	0.30 (0.14)	0.32 (0.18)	**0.006**

DKD, diabetic kidney disease; GLP-1, glucagon-like peptide-1; HDL-c, high-density lipoprotein cholesterol; LDL-c, low-density lipoprotein cholesterol; HbA1c, glycated hemoglobin; eGFR, estimated glomerular filtration rate; UACR, urinary albumin-to-creatinine ratio. Normally distributed continuous variables are presented as the means ± standard deviations, skewed distributed continuous variables are presented as medians (interquartile ranges), and categorical variables are presented as frequencies and percentages. Comparisons between the two groups (DKD group vs. Non-DKD group) were carried out via Student’s *t*-test for normally distributed variables, the Mann–Whitney U test for abnormally distributed variables, or Pearson’s chi-square test for categorical variables. *p* values < 0.05 are shown in bold.

**Table 2 biomedicines-12-02370-t002:** Multivariate analysis of the association between thyroid function and DKD.

Models	Model 1	*p* Value	Model 2	*p* Value	Model 3	*p* Value
T3	**0.519 (0.407−0.663)**	**<0.001**	**0.492 (0.369−0.655)**	**<0.001**	0.525 (0.874−1.322	0.525
T4	**0.943 (0.910−0.977)**	**0.001**	0.960 (0.922−1.000)	0.050	0.983 (0.928−1.040)	0.552
FT3	**0.789 (0.738−0.844)**	**<0.001**	**0.802 (0.742−0.867)**	**<0.001**	0.962 (0.877−1.055)	0.431
FT4	**0.926 (0.905−0.948)**	**<0.001**	**0.930 (0.906−0.955)**	**<0.001**	**0.961 (0.927−0.996)**	**0.030**

Multivariate logistic regression models were established to analyze the independent effect of thyroid function on the risk of diabetic kidney disease presented by odds ratios and confidence intervals, with Model 1 adjusted for age, sex, and BMI; Model 2 adjusted for age, sex, BMI, T2DM duration, and HbA1c levels; and Model 3 adjusted for age, sex, BMI, T2DM duration, HbA1c levels, systolic blood pressure, diastolic blood pressure, uric acid, total cholesterol, triglycerides, high-density lipoprotein cholesterol, low-density lipoprotein cholesterol, and serum albumin levels. T3, triiodothyronine; T4, thyroxine; FT3, free triiodothyronine; FT4, free thyroxine; BMI, body mass index; HbA1c, glycated hemoglobin; T2DM, type 2 diabetes mellitus. Bold indicates a significant *p* value < 0.05.

## Data Availability

The data that support the findings of this study are available from the corresponding author upon reasonable request.

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
