# Peer review of "Lower Free Thyroxine Levels Are Associated with Diabetic Kidney Disease in Males with Type 2 Diabetes Mellitus: An Observational Cross-Sectional Study"

_biomedicines, 2024, doi:10.3390/biomedicines12102370_

Round 1

Reviewer 1 Report

Comments and Suggestions for Authors

The description of the statistical analysis mentions that abnormally distributed variables are presented as medians and interquartile ranges, and between-group comparisons were performed using the Mann-Whitney U tests. However, Table 1 does not provide values for medians and interquartile ranges. Could the authors clarify why these values are not included?

Additionally, comparisons between the two groups (DN vs non-DN) in Table 1 were conducted using Student’s t-test, Mann-Whitney U-test, or Pearson’s Chi-squared test. Could the authors specify the circumstances under which the Mann-Whitney U-test was used?

Furthermore, what is the statistical significance of the variables GLP-1 analogues, HbA1c, Glycated albumin, and 24-hour urine output between the two groups (DN vs non-DN)?

It would be helpful if the authors included the Pearson correlation coefficient among these variables in Figure 2 or a supplementary table.

Comments on the Quality of English Language

The English language is clear and concise, but it could be refined for greater clarity and professionalism.

Reviewer 2 Report

Comments and Suggestions for Authors

The Authors aimed to explore the correlation between thyroid function and the development of diabetic nephropathy (DN) in patients with type 2 diabetes mellitus (T2DM).

The Authors concluded that „greater FT4 level was a protective factor against the development of DN in T2DM patients, especially in males with a cut-off value of 13.35 pmol/L (the area under the curve = 0.604)“.

The title of the article should be more specific, regarding that mentioned AUC for men barely exceeds 0.600, which is not the case for women (AUC=0.556).

-        The novelty of the study is questionable.

-        English editing is strongly recommended.

-        The Authors should provide the manufacturer’s name, town and country (in parentheses) for the biochemistry and immunochemistry analyzers for the determination of mentioned parameters.

-        The Authors should avoid the term „development“ DN since this was cross-sectional study.

-         Besides the version of the software, complete information on the manufacturer of the statistical software should be provided.

-        The accuracy of diagnostic test needs to be explained, since the area under the ROC curve (AUC) between 0.5 and 0.6 suggests bad accuracy of diagnostic test. AUC between 0.6 and 0.7 suggests sufficient accuracy, between 0.7 and 0.8 good accuracy, between 0.8 and 0.9 very good accuracy, whereas AUC higher than 0.9 suggests the excellent accuracy of diagnostic test (https://www.ncbi.nlm.nih.gov/pmc/articles/PMC4975285/)

-        The flow chart of the participants should be included.

-        The Discussion section could be improved by adding some information regarding the underlying features of abnormal thyroid function and metabolic alterations. In line with this, the Authors are referred to a recently published article: https://doi.org/10.3390/biomedicines12081862

Comments on the Quality of English Language

 Moderate editing of English language required.

Reviewer 3 Report

Comments and Suggestions for Authors

Major comments:

Line 70 : “ Development of DN” abruptly appeared from the results’ section. So, I recommend to clarify the definition and describe DN as development of DN in patients data collection section. 

Results: Table 1. Line 117-126

Is there independent clinical significance of urine creatinine concentration? That has significance if 24 h excretion (reflecting muscle volume) or calculate urine microalbumin/cr ratio.

Authors mentioned subclinical hypothyroidism in discussion section.  How many patients were applicable as subclinical hypothyroidism or low T3 syndrome in each group of this study?

Serum albumin (especially low range) affect thyroid hormones, Generally, it is lower in kidney disease population, so I recommend to add the table.

Multivariate analysis: In non-DKD definition,  UACR was lower than 30 mg/gCr from the beginning of study. Analysis of model 2 is more reasonable than model 3 which included UACR.  Is it appropriate to include UACR for multivariate analysis to compare Non-DN vs DN? 

Discussion

It is critical limitation of this study couldn't determine the causal relationship between thyroid hormone levels and the development of DN.

 Duration of DKD was not unclear in the development of DN because of cross-sectional study. However, if this variate is impossible to include for baseline, it becomes limitation of this study.

Minor comments:

Line 92-

... serum creatinine, cystatin C are correct.

What are the methods of analysis for urine sample ?

What is the formula for estimated GFR?

Results:

Line117 Description of 2 groups is different from table 1.

I think distribution of eGFR and albuminuria are important to understand the thyroid hormone marker in DN subjects.

I recommend to add to distribution of CKD classification by eGFR and alubuminuria,  or classification of DKD by stage 2(without albuminuria and eGFR=>30, stage 3 (with albuminuria and eGFR over 30), and stage 4 (eGFR<30 ) of total DN subjects.

Table 1. Line 117-126 :Is there independent clinical significance of urine creatinine concentration?  That has significance if 24 h excretion (reflecting muscle volume) or calculate microalbuminuria/cr ratio.

The unit of UACR “mg/g” should be corrected to mg/gCr .

Reviewer 4 Report

Comments and Suggestions for Authors

This study aimed to explore the correlation between thyroid function and the development of diabetic nephropathy (DN) in patients with type 2 diabetes mellitus (T2DM) through a cross-sectional study. However, I have some concerns and queries regarding the article that I would like to articulate:

1.Similar literature has been reported previously, and this paper lacks innovation. 

2.Inclusion and exclusion criteria should be strictly controlled to reduce confounding factors. For example, conditions such as a history of thyroid disease or taking any drugs that affect thyroid function should be included as one of the exclusion criteria.

3. As mentioned in the paper, blood pressure, blood glucose and blood lipid may be involved in the development of diabetic nephropathy. In Table 1, it is suggested to add clinical characteristics such as lifestyle (smoking and drinking), family history and blood lipid.

4. It is recommended to add the results of P value in Table 2, and in the Model 1, the effect of FT3 is better than that of FT4, and it is recommended to enrich the description of the results.

5. In Figure 3, the AUC is small and the accuracy is low, indicating low clinical value.

6. The literature referenced by the author is old, and it is suggested to update it.

Comments on the Quality of English Language

Many parts of the manuscript are literal translations in Chinese, and appropriate English polishing is recommended.

Round 2

Reviewer 1 Report

Comments and Suggestions for Authors

The authors have addressed all of my comments, and I have no additional comments.

Author Response

Thank you for your review!

Reviewer 2 Report

Comments and Suggestions for Authors

The Authors have made corrections according to the Reviewer's suggestions.

Author Response

Thank you for your review!

Reviewer 3 Report

Comments and Suggestions for Authors

The manuscript has been revised well. Thank you for your massive effort to improve the article.

Author Response

Thank you for your review!